# Metataxonomic Mapping of the Microbial Diversity of Irish and Eastern Mediterranean Cheeses

**DOI:** 10.3390/foods11162483

**Published:** 2022-08-17

**Authors:** Eleni Kamilari, Dimitrios Tsaltas, Catherine Stanton, R. Paul Ross

**Affiliations:** 1APC Microbiome Ireland, University College Cork, T12 YT20 Cork, Ireland or; 2School of Microbiology, University College Cork, T12 K8AF Cork, Ireland; 3Department of Agricultural Sciences, Biotechnology and Food Science, Cyprus University of Technology, Lemesos 3036, Cyprus; 4Department of Biosciences, Teagasc Food Research Centre, Moorepark, Fermoy, Co., P61 C996 Cork, Ireland

**Keywords:** cheese, microbiome, 16S rDNA, ITS loci, high throughput sequencing, ripened cheeses, biomarkers, lactic acid bacteria, starter cultures, metataxonomic sequencing

## Abstract

The distinct sensorial characteristics of local cheeses influence consumer preferences, and make an essential contribution to the local economy. Microbial diversity in cheese is among the fundamental contributors to sensorial and qualitative characteristics. However, knowledge regarding the existence of microbial patterns associated with regional production practices in ripened cheeses remains limited. The present research was conducted to test the hypothesis that the background metagenome of cheeses could be used as a marker of their origin. We compared Irish versus Eastern Mediterranean cheeses—namely Greek and Cypriot—using High Throughput Sequencing (HTS). The study identified a significantly distinct separation among cheeses originating from the three different countries, in terms of the total microbial community composition. The use of machine learning and biomarkers discovery algorithms defined key microbes that differentiate each geographic region. Finally, the development of interaction networks revealed that the key species developed mostly negative interactions with the other members of the communities, highlighting their dominance in the community. The findings of the present research demonstrate that metagenome could indeed be used as a biological marker of the origin of mature cheeses, and could provide further insight into the dynamics of microbial community composition in ripened cheeses.

## 1. Introduction

The dairy industry is an essential contributor to the Irish and East Mediterranean economies, with cheese production being a principal part of the regional gastronomical cultures. The unique sensorial characteristics of regional cheeses arise from a combination of factors, including: (1) the type of milk used in cheese production; (2) cheese manufacturing practices, such as the addition of natural whey or starter cultures, salt, herbs or other ingredients; (3) whether the milk is raw or pasteurized and (4) the degree of ripening [1,2]. During cheese ripening, complex and dynamic interactions take place among the members of the microbial community. The specific microbial patterns that are developed as a result of these interactions influence the taste and flavor, and affect the quality of the final product [3]. The use of pasteurized milk and defined starter cultures simplifies the composition and the dynamics of the microbial communities. Nevertheless, the microbial associations and the effect produced by the community metabolites on cheese texture, flavor, aroma and other qualitative characteristics remain puzzling. Despite the fact that several factors contribute to microbial diversity shaping, the existence of region-associated microbial signatures has been revealed by recent metataxonomic sequencing studies [4,5].

The advent of High Throughput Sequencing (HTS) technologies has remarkably improved our understanding of the role that microbial consortia play in the process of cheese ripening [1,6,7]. The composition of the bacterial communities is comprised of the following: the dominants [the lactic acid bacteria (LAB)], which are the most successful fermenters of the configured microenvironment; the subdominants, which are the survivors of the selective pressure of the dominants, and exist in lower relative abundance; and the very low abundant species, the presence of which is attributed to environmental contamination [8]. The metabolic activity of specific LAB, including *Lactococcus lactis* and *Streptococcus thermophilus*, to successfully ferment lactose, favors their use as starter cultures to guide the process of fermentation, and to affect cheese microbiome shaping and sensorial characteristic development [9]. However, during the process of ripening, the microbiota can become dominated by Non-Starter Lactic Acid Bacteria (NSLAB) [8]. These species contain increased proteolytic and lipolyticactivities. Their ability to utilize other carbon sources besides lactose results in different flavors and textures in cheeses. Apart from LAB, cheese microbiota can be comprised of spoilage microorganisms and pathogens, including the genera *Clostridium*, *Staphylococcus*, *Bacillus*, *Pseudomonas*, *Enterobacter*, and other Proteobacteria. Moreover, yeast including *Candida* spp., *Debaryomyces hansenii*, *Yarrowia lipolytica*, and *Kluyveromyces lactis*, as well as molds including *Aspergillus*, *Geotrichum*, *Penicillium*, *Fusarium*, and *Mucor* are found among cheese microbial consortia [10].

Cheese spoilage and foodborne infections constitute major challenges for the cheese industry and cheese producers, and are of great concern for consumers [11]. Cheese storage under refrigeration conditions favors the development of psychrotrophic bacteria, including *Pseudomonas* spp., *Acinetobacter* spp., *Bacillus* spp., and *Enterobacteriaceae* [12,13], which can release great amounts of extracellular heat-stable hydrolytic enzymes [14]. Their production results in spoilage, and degradation of the quality of cheese, by creating rancid, or bitter flavors. Proteases break down casein, leading to bitterness in milk, gel formation of UHT-sterilized milk, and reduction of the amount of soft cheese [15]. Lipases degrade triglycerides to glycerol and free fatty acids, producing off-flavors [16]. Moreover, lecithinases lead to the disruption of milk lipid globule membranes, enhancing the catalytic activity of lipases [16]. Elevation of the pH > 5.0 during cheese ripening supports the growth of coliforms, which can produce acid and gas [17]. Heterofermentative LAB, including *Lactobacillus* and *Leuconostoc* secrete gas and off-flavors, promoting cheese spoilage [18]. Fungi produce a plethora of secondary metabolites, and some of these, such as off-odors and flavors, have a negative influence on the organoleptic characteristics of mature cheese [19].

In the present study, metataxonomic sequencing analysis was applied to characterize the microbial communities of mature Irish and eastern Mediterranean cheeses, and to map the microbial signatures associated with each region. Considering that several factors affect cheeses’ microbial diversity composition, our goal was to identify the key biomarkers associated with each geographic location, and to reveal critical interactions among the members of the communities. Our findings may enhance existing knowledge of the microbial community dynamics of ripened cheeses, and benefit local cheese producers by supporting the improvement of product quality.

## 2. Materials and Methods

### 2.1. Sample Collection

Cheeses were collected between March and May 2021, and included 16 mature cheeses from the Cork (Ireland) market, 10 from markets and producers in Cyprus, 10 from Crete (Greece), 10 from Mytillini (Greece), and 3 from Limnos (Greece) (Figure 1, Table 1). Cypriot and Greek cheeses were placed in cool boxes, and transported to University College, Cork within 3 working days, whereas Irish cheeses were transferred immediately to University College, Cork. Samples were stored at −20 °C until processing.

### 2.2. Metagenomic DNA Extraction

For sample homogenization, 5 g of cheese (only the core, not the rind) were mixed with 45 mL of 2% tri-sodium citrate (Honeywell, Europe) using a Stomacher 400 Circulator (Seward, UK) at 300 rpm for 2 minutes. After centrifugation at 16,000× *g* for 5 min at 4 °C, the top fat layer was removed using sterile cotton swabs, the supernatant was discarded, and the pellet was used for DNA extraction. Microbial DNA extraction was performed using DNeasy^®^ PowerFood^®^ Microbial Kit (MoBio Laboratories Inc., Carlsbad, CA, USA) according to the manufacturer’s instructions, with the following modification: after the addition of 450 µL Solution MBL during the cell lysis step, the samples were incubated for 10 min at 65 °C, and for an additional 10 min at 95 °C. The extracted DNA was stored at −20 °C until processing.

### 2.3. Quantification of Total DNA

The total DNA isolated from the cheese samples was quantified fluorometrically with Qubit 4.0 fluorometer (Invitrogen, Carlsbad, CA, USA), using Qubit dsDNA HS Assay Kit (Invitrogen). The purity of the DNA was evaluated by measuring the ratios of absorbance, A260/280 nm and A260/230 nm, using a spectrophotometer (NanoDrop Thermo Scientific, Wilmington, DE, USA).

### 2.4. Barcoded Illumina MiSeq Amplicon Sequencing of Bacterial 16s rRNA Gene and of Fungal ITS Region

The 16S rRNA bacterial gene amplification, the Illumina paired-completion library preparation, and sequencing were performed as described previously [20]. The 16S rRNA bacterial gene was amplified using the primers V3: 5′-TCGTCGGCAGCGTCAGATGTGTATAAGAGACAG-3′ and V4: 5′-GTCTCGTGGGCTCGGAGATGTGTATAAGAGACAG-3′, whereas the fungal internal transcribed spacer 1 (ITS1) locus was amplified using the primers BITS (5′-NNNNNNNNCTACCTGCGGARGGATCA-3′) and B58S3 (5′-GAGATCCRTTGYTRAAAGTT-3′), with the addition of the overhang adapter sequence, according to Bokulich and colleagues [21]. The fungal ITS1 locus amplification and sequencing, PCR amplicon purification, evaluation of DNA quantity and quality, and amplicon normalization were also conducted as described previously [22,23]. For the sequencing run, both the bacterial 16S rRNA gene and the ITS1 loci libraries were loaded on MiSeq 600 cycle Reagent Kit v3 (Illumina, San Diego, CA, USA) (5% PhiX) and run on a MiSeq Illumina sequencing platform.

### 2.5. Microbiome and Statistical Analysis

Raw fastq sequences were quality filtered, and Qiime 2 version 2020.2 was used to calculate the diversity indexes Shannon, Simpson, and Chao1, regarding alpha diversity; unweighted [24] and weighted [25] uniFrac distances, regarding beta diversity; and observed OTU diversity metrics [26], as previously described [27]. Samples were rarefied at 15,637 for bacteria and 19,145 for fungi. To visualize the distinction of the microbial communities into clusters based on Unweighted UniFrac distances, Principal Coordinate Analysis (PCoA) was performed using q2-diversity after the samples were rarefied (subsampled without replacement) with 25 sampling depths. The significance among Irish, Greek, and Cypriot cheeses was tested with PERMANOVA [28] in Qiime 2 with 999 permutations (Least Significant Difference (LSD) *p* ≤ 0.05). Taxonomy assignment for 16S rDNA sequences into OTUs was based on the q2-feature-classifier [29] against the Greengenes 13_8 99% OTUs reference sequences [30], whereas for fungal ITS loci the UNITE fungal internal transcribed spacer (ITS) database (9_12 release) [31] was applied. Sequence filtering was applied to remove taxonomies with unsuccessful identification to phylum level. Cheeses with more than 60% of unidentified genera, including C10, I1, and I16 for fungi, were excluded from downstream analysis. To discover biomarkers for different origin cheeses, the LEfSe algorithm was used [32], as described previously [28]. Machine learning was applied in the q2-sample-classifier, using the Random Forest algorithm in Qiime2, as described by Bokulich and colleagues [33], and Pedregosa and colleagues [34]. Receiver operating characteristic (ROC) curves, Per-Class Receiver Operating Characteristics (PCROC), area under the curve (AUC), feature importance scores, and an abundance heatmap representing the most important features in each sample/group were created as part of the q2-sample-classifier pipeline (using default parameters). Calculation of AUC was performed using scikit-learn [34] for each class, and micro- and macro-averages. Micro-average was evaluated throughout every sample, and influenced by class imbalances. Macro-average evaluates equally the classification of every sample, diminishing the effect of class imbalances on the average AUC. Strong (r > 0.6 or r < −0.6) and significant (*p* < 0.01) correlations between the identified microbial taxa from all cheese samples were evaluated using Pearson and Spearman correlations and dissimilarities using Bray Curtis and Kullback–Leibler matrices in CoNet [35], and a co-occurrence network was created using Cytoscape 3.2.1 [36]. Taxa with sum relative abundances less than <0.01% [37] appeared in less than 20% [38].

All raw sequence data in read-pairs format were deposited in the National Centre for Biotechnology Information (NCBI) in the Sequence Read Archive (SRA) under the Bioproject PRJNA838921.

## 3. Results

### 3.1. Abundance and Alpha Diversity of the Cheese Microbiota

Mature cheese samples collected from Ireland, Greece, and Cyprus were each analyzed for their bacterial and fungal diversities. To estimate the bacterial diversity of cheeses, 48 examined samples were sequenced to generate 1,840,004 high-quality sequencing reads, with an average of 38,333.42 sequencing reads per sample (range = 15,637–125,662, STD = 21,614.82; Appendix A). High-quality sequences were grouped into the average number of 57 OTUs (range = 24–108, SD = 21; Appendix A). To evaluate the fungal diversity of mature cheeses, 49 examined samples were used as input to the Illumina MiSeq to produce 4,401,335 high-quality sequencing reads, with an average of 89,823.16 sequencing reads per sample (range = 12,493–337,284, STD = 84,393.48; Appendix A). High-quality sequences were grouped into an average of 82.79 OTUs (range = 12–256, SD = 50.4; Appendix A). Shannon, Simpson, and Chao1 estimators are shown in Appendix A for bacterial and fungal diversities, respectively.

Initially, we evaluated the microbial alpha diversity based on the Shannon, Simpson, and Chao1 estimators. Regarding the bacterial alpha diversity, a significant difference (*p* = 0.002) was observed between the Irish and Greek cheeses, based on the Chao1 index, using the Kruskal–Wallis test. The Cypriot cheeses indicated higher fungal alpha diversity, but the difference was not significant based on the Kruskal–Wallis test (Figure 2B, Appendix A). No other significant difference was observed in the microbial alpha diversity among the cheeses from the three countries.

### 3.2. Microbial Beta Biodiversity among the Different Regions

Beta-diversity analysis using an Unweighted/Weighted Unifrac metric was performed, to qualitatively estimate the presence of distinct microbial patterns among cheeses produced in Ireland, Greece, and Cyprus. Both the bacterial and fungal diversities of the Irish cheeses were grouped separately from the Greek and Cypriot cheeses on the PCoA plot (Figure 3A,C (data presented by country), Figure 3B,D (data presented by town/island)). Regarding the bacterial diversity, the PERMANOVA test indicated a significant difference between Irish and Greek, and Irish and Cypriot cheeses (*p* < 0.05; Appendix A), based on both the Unweighted and the Weighted Unifrac metrices. A significant difference in the fungal beta diversity of the cheeses was observed among all three countries (*p* < 0.05; Appendix A), based on the Unweighted Unifrac metric, but only between cheeses from Ireland and Greece based on the Weighted Unifrac metric.

### 3.3. Taxonomic Composition of Microbial Communities

In general, the dominant bacterial genera in analyzed cheeses were *Lactococcus*, *Streptococcus*, and *Lactobacillus* (Figure 4A,B). All Irish cheeses, except for two produced from raw milk, were dominated by *Lactococcus*. Greek cheeses were mostly dominated by *Streptococcus* sp. and *Lactobacillus* spp., and a few by members of the genus *Lactococcus*. *Streptococcus*, followed by *Lactobacillus*, were the predominant genera in all Cypriot cheeses. Greek and Cypriot cheeses were characterized by increased relative representation of the species *Lactobacillus helveticus*. These species were also detected in some Irish cheeses. Mostly Cypriot cheeses showed increased relative representation of the species *Lactobacillus delbrueckii*. Some cheeses from all countries indicated increased relative abundance of the species *Lactobacillus zeae* and *Lactobacillus brevis*. Apart from the aforementioned LAB, some cheeses indicated increased relative representation of the genus *Leuconostoc*, and some cheeses of the species *L. mesenteroides*, but in lower relative abundance.

Regarding the lower relative abundance bacteria, the spoilage family *Enterobacteriaceae* and the genus *Staphylococcus* were detected. However, three cheeses from Ireland indicated increased relative abundance (5%, 9%, and 33%) of *Staphylococcus* species, including *S. equorum*. Additionally, some Greek cheeses, mostly from Crete and Limnos, had increased relative representation of the species *S. equi*. Some Irish cheeses were characterized by the presence of members of the families *Micrococcaceae* and *Peptostreptococcaceae*, of the genera *Psychrobacter*, and *Sphingomonas*, represented by the species *S. yabuuchiae*, and of the species *Vibrio rumoiensis*. Greek cheeses showed low relative abundances of the families *Bifidobacteriaceae* and *Weeksellaceae*, including *Wautersiella* that were detected in cheeses from Mytillini, but were also characterized by members of the genera *Enterococcus*, *Acinetobacter*, *Kocuria*, and *Macrococcus*, and of the species *Kurthia gibsonii*. Regarding the genus *Acinetobacter*, the most commonly detected species included *A. johnsonii* and *A. rhizosphaerae*. The species *A. guillouiae* was detected only in cheeses from Mytillini. Cheeses from Mytillini were also characterized by the family *Bacillaceae*, by the genus *Fructobacillus*, and by the species *Corynebacterium variabile*. Two cheeses from Crete indicated the presence of the species *Clostridium perfringens*, a species which is known to cause food poisoning [39]. Cypriot cheeses showed the presence of the contaminant *Mycoplasma*, with three cheeses having relative representations of 2.8%, 5.5%, and 9.4%. Moreover, the analysis revealed the co-occurrence of some bacterial taxa in low relative abundances in cheeses from Greece and Cyprus. Specifically, cheeses from both countries had representatives of the species *Pseudomonas*, *Chryseobacterium*, *Enhydrobacter*, and *Pediococcus*. *Pseudomonas* was also detected in an Irish cheese. *Pediococcus* characterized mostly cheeses from Crete. Cheeses from Cyprus and Crete also showed the presence of *Chromohalobacter*.

The fungal diversity of all the analyzed mature cheeses, according to ITS1 loci sequencing, was dominated by members of the order Saccharomycetales and, in lower relative abundances, of the order Eurotiales of the phylum Ascomycota. Some cheeses had representatives of the orders Mucorales, Capnodiales, Trichosporonales, Hypocreales, and Pleosporales. The most abundant families were *Debaryomycetaceae*, *Aspergillaceae*, *Saccharomycetaceae*, and *Saccharomycodaceae*, and in lesser amounts, *Mucoraceae*, *Cladosporiaceae*, and *Trichosporonaceae*. The common and abundant identified fungal genera in most analyzed cheeses were: *Debaryomyces*, represented by the species *D. hansenii*; *Penicillium*, represented by the species *P. carneum*; and *P. commune*, and *Candida*, represented mostly by the species *C. zeylanoides*, *C. parapsilosis*, and *C. inconspicua* (Figure 5A,B). Specifically, the species *P. carneum* was predominant in the Irish cheeses I2 and I9, in which *P. roquefortii* was used as a starter, indicating that these species might have been misidentified due to sequence similarities. The genus *Kluyveromyces* was also predominant in Greek and Cypriot cheeses, represented by the species *K. marxianus* and *K. lactis*.

Members of the genera *Saccharomyces*, such as *S. paradoxus*, *Aspergillus*, such as *A. westerdijkiae*, and *Trichosporon*, such as *T. asahii*, represented the lower abundant species in most analyzed samples. Some cheeses were characterized by the presence of the genera *Issatchenkia*, such as *I. orientalis*, *Kazachstania*, *Torulaspora*, *Yamadazyma*, *Malassezia*, and *Rhodotorula*. The species *Cutaneotrichosporon curvatus* was detected in very low relative abundance (>1%) in many cheeses, apart from cheeses from Crete that were detected in relative abundance from 1–6,5%. Greek and Cyprus cheeses had representatives of the genera *Hanseniaspora*, such as *H. nectarophila* and *H. guilliermondii*, *Alternaria*, such as *A. alternata*, *Aureobasidium*, such as *A. pullulans*, *Meyerozyma*, *Stemphylium*, *Botrytis*, *Mycosphaerella*, *Wickerhamomyces*, and *Sporobolomyces*. The species *Parengyodontium album* and *Erysiphe necator* were detected in cheeses from Limnos and Mytillini, and one Cypriot cheese. Moreover, the genus *Dipodascus* was detected in cheeses from Ireland and Greece, but not from Cyprus. Finally, members of the genera *Mycena* and *Dekkera* were detected only in cheeses from Cyprus, and the species *Candida glaebosa* only in Irish cheeses. Notably, a high percentage of the ITS1 loci raw reads could not be classified at the genus level.

### 3.4. Identification of Microbial Biomarkers

To evaluate whether the values of the identified relative abundances of microbial taxa were differentially distributed among the cheeses from the three countries, we applied the LEfSe algorithm. The analysis revealed that species of the detected bacterial biomarkers are applied by the industry as starter cultures (Figure 6). Specifically, Irish cheeses had a significantly increased relative representation of members of the genus *Lactococcus* compared to cheeses from Cyprus and Greece (Figure 6A). Greek cheeses indicated a significantly more abundant representation of the genus *Streptococcus*, and in cheeses from Cyprus of the species *L. delbrueckii*. Apart from LAB, cheeses from Greece had significantly increased representation of the genera *Staphylococcus*, *Chryseobacterium*, and *Enhydrobacter*.

Regarding fungi, Irish cheeses had increased representation of the species *Penicillium carneum* and *Candida glaebosa* (Figure 6B). Greek cheeses indicated increased relative representation of the genera *Botrytis* and *Filobasidium*, and of the species *Kluyveromyces marxianus*, *Kluyveromyces lactis*, *Erysiphe necator*, *Meyerozyma guilliermondii*, *Issatchenkia orientalis*, *Saccharomyces paradoxus*, *Candida parapsilosis*, *Leptobacillium leptobactrum*, *Parengyodontium album*, and *Trichosporon asahii* (Figure 6B, Appendix A). Additionally, cheeses from Mytillini were mostly associated with *Mycosphaerella tassiana*, Limnos cheeses with *Aureobasidium pullulans*, and Crete cheeses with *Cutaneotrichosporon curvatus* (Appendix A). Finally, the taxa detected in increased relative abundance in Cypriot cheeses included the genera Pichia and Hanseniaspora, and the species *Hanseniaspora guilliermondii*, *Hanseniaspora nectarophila*, *Candida stellata*, and *Malassezia restricta* (Figure 6B).

### 3.5. Differentiation of Cheeses from Each Country with Great Accuracy, Using Random Forests Classifier

A supervised learning Random Forest classifier was trained to predict, using the bacterial 16S rRNA gene and the ITS loci OTU abundance data, the country of origin of each cheese based on the detected microbiota. The results indicated a very high predictive accuracy (average AUC = 1.00 regarding bacteria, and average AUC = 0.98 regarding fungi: Figure 7A,C, respectively), showing that the algorithm could predict the origin of key members of the cheese microbiota with high accuracy. Most of the top 13 predictive features that were identified for the bacteria were in agreement with the results obtained using the LefSe algorithm (Figure 7B). Specifically, *Lactococcus* differentiated Irish cheeses, *Streptococcus* and *Staphylococcus* differentiated Greek cheeses, and *L. delbrueckii* differentiated Cypriot cheeses. In addition, *L. brevis* characterized mostly Irish cheeses, *Mycoplasma* and *Leuconostocaceae* characterized Cypriot cheeses, and *Chryseobacterium*, *Enhydrobacter*, *Pseudomonas*, and *Streptococcus equi* characterized Greek cheeses, but with reduced detected frequency.

Similarly, the majority of the top 30 predictive features for fungi corresponded to those identified by the LefSe algorithm biomarkers (Figure 7D). The species *D. hansenii*, *C. glaebosa*, and *P. carneum* differentiated mostly Irish cheeses. The species *K. marxianus*, *K. lactis*, *S. paradoxus*, and *I. orientalis* distinguished Greek cheeses and, to a lesser degree, the species *P. album*, *A. pullulans*, *C. parapsilosis*, *I. orientalis*, *T. asahii*, and *C. curvatus*. Finally, *H. nectarophila* differentiated Cyprus cheeses.

### 3.6. Microbial Network Analysis

To study the correlations between the identified bacterial and fungal taxa for each country, a co-occurrence network was constructed using the CoNet algorithm in Cytoscape, based on the Pearson and Spearman correlations and the Bray Curtis and Kullback–Leibler dissimilarity matrices (Figure 8). The analysis focused on the keystone microbes that were identified as biomarkers strongly associated with each geographic region. In general, the detected and abundant bacterial biomarkers developed negative associations with the other members of the cheese microbiota. Specifically, in Irish cheeses, *Lactococcus* was negatively associated with *Streptococcus*, *L. zeae*, *Brevibacterium*, *C. parapsilosis*, and *D. hansenii* (Appendix A). Likewise, in Greek cheeses *Streptococcus* indicated negative associations with *Lactococcus* (Appendix A). Additionally, *Streptococcus* sp. were negatively associated with *Lactobacillus*, *Leuconostoc*, *P. carneum*, and *Malassezia globosa*. Moreover, *L. delbrueckii*, the species that differentiated Cypriot cheeses, was negatively correlated with *Lactococcus*, *Pseudomonas*, and *D. hansenii*. However, *L. delbrueckii* created a co-occurrence network with *Streptococcus*, *Hanseniaspora* spp., *M. restricta*, and *Penicillium* (Appendix A).

Apart from *Lactococcus*, *D. hansenii*, the predominant fungi in most Irish cheeses was negatively associated with *L. helveticus*, *K. marxianus*, *P. carneum*, *C. glaebosa*, and *C. inconspicua* (Appendix A). Additional negative correlations were observed between *K. marxianus* and *L. brevis*, *Chromohalobacter*, *Pseudomonas*, *D. hansenii*, *C. stellata*, *C. magnoliae*, *M. globosa*, *H. vineae*, *L. leptobactrum*, and *P. album* in Greek cheeses (Appendix A). *K. lactis* was also negatively associated both with *L. brevis*; and with some spoilage microbes, including *L. mesenteroides*, *Enterococcus*, *Klebsiella*, *Acinetobacter* spp., and several other fungi. Finally, in Cypriot cheeses, *Hanseniaspora* spp. were negatively correlated with *Lactococcus*, *Pseudomonas*, and *Penicillium* spp., and positively correlated with *M. restricta* and *Mycoplasma* (Appendix A).

## 4. Discussion

Substantial evidence indicates the existence of distinguishable microbial patterns in ripened cheeses originating from different areas and produced using different manufacturing conditions [1,4,40,41]. Apart from the cheese manufacturing environment, factors including the origin of the milk, the addition of starter cultures or natural whey, herbs, salt, and other ingredients, and the processing conditions, such as milk heating or pasteurization, influence the shaping of unique microbial consortia in each type of cheese [1]. In the present study, the microbial diversity of mature Irish and eastern Mediterranean cheeses was analyzed using metataxonomic sequencing, to reveal the presence of common microbial patterns associated with each location. Moreover, we suggest microbial biomarkers related to the three areas, and provide insight into the dynamics of the microbial communities’ compositions.

Initially, we demonstrated that phylogeny-based beta-diversity, as depicted by UniFrac PCoA and confirmed by PERMANOVA test, could separate the microbial diversity of cheeses from the three countries, Ireland, Cyprus and Greece, into different cluster affinities. To further strengthen our results, we used a Random Forest algorithm that predicted, with strong predictive accuracy, the origin of the cheeses based on the microbiota. These findings are in line with similar metataxonomic sequencing-based studies, revealing a microbiological mapping associated with each cheese’s geographical origin and way of production [4,42]. Specifically, cheeses from all countries indicated a significant difference in fungal beta diversity. Additionally, the bacterial diversity of Irish cheeses was significantly distinct from Greek and Cypriot cheeses. Kamimura and colleagues [4] performed a metataxonomic analysis of 11 different types of cheeses originating from five geographical regions of Brazil, and likewise identified microbial signatures associated with the area of production. Bokulich and Mills [42] indicated that washed-rind mature cheeses’ microbial diversity was dominated by environmental microbes found in processing environments. However, Wolfe and colleagues [43] analyzed cheese rind microbial communities from 10 countries, to reveal that the microbial community compositions were mostly associated with abiotic factors, such as moisture, rather than geography, highlighting the effect of the processing conditions on the cheese microbiota formation.

The metataxonomic data, as applied to the Random Forest and the LEfSe biomarkers discovery algorithms, identified distinct microbiological signatures among the cheeses from the three different countries. Specifically, regarding the bacterial community composition, *Lactococcus* was mostly associated with Irish, *Staphylococcus* with Greek, and *L. delbrueckii* with Cypriot cheeses. Notably, the species *Lactococcus lactis*, *L. delbrueckii*, and *Streptococcus thermophilus*, were applied by the industry as starter cultures [44,45,46,47,48]. The predominant presence of *Lactococcus* sp. in Irish cheese was in agreement with Walsh and colleagues [49], and with Quigley and colleagues [50]. These studies also identified the presence of *Leuconostoc*, *S. thermophilus*, *L. helveticus*, *S. equorum*, and *Psychrobacter* in lower relative abundances. Michailidou and colleagues [51] analyzed 6 Greek PDO (Protected Designation of Origin) cheeses from 3 different producers or industries, using amplicon metabarcoding analysis, and found that most cheeses were dominated by species that belonged to the genus *Lactococcus*, while the rest were co-dominated by *Streptococcus* spp. and *Lactobacillus* spp., which was similar to our own findings. However, in our analysis, *Streptococcus* spp. was predominant in most cheeses. This may have been due to the fact that, apart from Kalathaki Limnou, those cited above were analyzing cheeses originating from locations different to the ones that we analyzed. Moreover, metagenomic analysis of Greek feta PDO cheeses indicated that the microbiota was dominated by the species *L. lactis*, whereas *Streptococcus* and *Lactobacillus* were detected in lower relative abundance [52]. Kamilari and colleagues [20] analyzed the bacterial diversity of Halloumi cheese, a semi-hard cheese produced in Cyprus, to find that the dominant genus was *Lactobacillus*. Papademas and colleagues [47] performed a metagenomic analysis on the microbiota of Halitzia cheese, a traditional, white-brined cheese produced in Tilliria of Paphos, to identify *Lactococcus*, *Lactobacillus*, and *Leuconostoc* as the predominant genera. These results show that differences in the manufacturing conditions, including the degree of ripening, have a critical effect on the cheese microbiota formation. In agreement with this finding, metataxonomic analysis of the French “Tomme d’Orchies” cheese indicated a drastic reduction in the relative abundance of *Lactococcus* and an increment of *Streptococcus* spp. during the ripening period [53].

To identify how the aforementioned bacterial biomarkers influenced the microbial community dynamics, we constructed a co-occurrence network among the detected bacterial and fungal taxa per country, focusing on the detected biomarkers with their neighbor taxa associations. The analysis revealed that these key microbes were mostly negatively associated with the other members of the cheese microbiota. The predominance of *L. lactis* in ripened cheeses has been shown in several studies [45,54,55]. Their ability to survive in hard cheeses is due to shifting their metabolism to peptide and amino acid consumption [45], and their predominance is possibly due to antimicrobial peptide and bacteriocin production [56,57]. Similarly, *Streptococcus* and, specifically, *S. thermophilus* have the capability to consume amino acids producing aromatic compounds in hard cheeses [58]. This, in combination with their possession of genes encoding for bacteriocin and exopolysaccharide biosynthesis, may provide them with the requisite advantage to dominate over the other microbes in hard cheeses [59,60]. Likewise, *L. delbrueckii* possesses genes encoding enzymes for cheese proteolysis, exopolysaccharide, and bacteriocin production [61,62]. In addition to the detected negative associations, we also found positive associations between *L. delbrueckii* and *Streptococcus*. These results are in agreement with Charlet and colleagues, [63], and Parente and colleagues [3], indicating that these interactions are critical for their co-administration as starters in cheese-making, to prevent pathogenic or spoilage bacteria growth.

Apart from the predominant LAB, some spoilage bacteria were detected in low relative abundance in most cheeses. Specifically, some cheeses had members of the family *Enterobacteriaceae* and the genus *Staphylococcus*. The family *Enterobacteriaceae* includes some severe foodborne pathogens, including toxin producing *Shigella* spp., *Salmonella* spp., and some serotypes of *E. coli*, such as *E. coli* O157:H7 [64]. Among the *Staphylococcus* spp., *S. aureus*, which is commonly detected in cheeses, is an important foodborne pathogen and disease-causing bacterium in humans [65]. Moreover, two cheeses had low relative abundance of the species *C. perfringens*. *C. perfringens* is responsible for several enterotoxic and histotoxic diseases in humans, caused by the protein toxins it produces [66]. These results indicate that the application of post-pasteurization steps needs to be considered by cheese producers, to ensure the absence of these contaminants in cheeses.

ITS loci sequencing indicated that most of the analyzed cheeses were characterized by an increased relative representation of the species *D. hansenii*. The dominant presence of this species in cheese microbiota was detected in previous metataxonomic sequencing studies in Irish [49], Greek [51], and Cypriot cheeses [47]. The ability of *D. hansenii* to grow in elevated NaCl concentrations, with lactate as the basic carbon source, and low pH, favors their growth in cheeses during ripening [67]. Additionally, some Irish cheeses, two Greek, and two Cypriot cheeses were characterized by an increased relative abundance of the species identified as *P. carneum*. These species are closely related to *Penicillium roqueforti*, a species that is used as a starter for blue cheese production (Table 1), and was formerly classified as a variety of *P. roqueforti* [68]. However, in contrast to *P. roqueforti*, *P. carneum* produces a mycotoxin called patulin, which is harmful to human health. Due to their high ITS loci sequence similarity, *P. roqueforti* might have been misidentified as *P. carneum*. Notably, in addition to blue cheeses, *P. carneum* has also been detected in other cheeses. In addition to the aforementioned, *P. commune*, *C. parapsilosis*, *C. zeylanoides*, and *C. inconspicua* were among the predominant detected species. These spoilage fungi are among the most commonly occurring in cheeses [69,70]. *P. commune* is among the predominant species in low water content cheeses. Irish cheeses were characterized by the presence of the species *C. glaebosa*. These species have also been detected in other hard cheeses, such as Pecorino di Farindolais, [71], and cheeses from the province of Quebec, Canada [10], especially at the beginning of ripening. Greek cheeses showed an increased relative representation of the species *K. marxianus* and *K. lactis*, in line with Michailidou and colleagues [51]. The predominance of these *Kluyveromyces* species in ripened cheeses is due to their ability to ferment lactose, but they also possess proteolytic and lipolytic abilities [72]. Moreover, Cypriot cheeses showed an elevated relative abundance of the *Hanseniaspora* species. The presence of *Hanseniaspora* species is rarely detected in cheese samples [73]; their contribution to cheeses’ sensorial and qualitative characteristics therefore remains unknown, and this warrants further investigation.

In the future, more mature cheeses originating from different cities and countries, and produced not only from pasteurized, but from raw milk, will be analyzed and compared with similar types of cheeses, using an amplicon sequencing approach. Data will be added in dedicated food microbiome databases. This will allow us to identify biomarkers that might define PDO cheeses’ authenticity, and differentiate them from fraudulent products that may exist in the market. To achieve this, we will combine metataxonomic sequencing and Shotgun sequencing, with metabolomics, to discover the influence of the microbial consortia on cheeses’ aromatic profiles and flavors.

## 5. Conclusions

The present metataxonomic research provides a snapshot of the microbiological mapping of mature cheeses produced in different geographic locations. The study identified the microbial communities in several mature Irish, Greek, and Cypriot cheeses. Furthermore, it revealed the presence of key microbial species that differentiate Irish, Greek, and Cypriot mature cheeses. Even though the detected bacteria were those that were possibly used as starters, the development of interaction networks increased our understanding of how these microbes might affect community composition. Finally, the presence of several spoilage and potential pathogenic microbes raises a concern about the need for additional post-pasteurization measures in production of some mature cheeses. The findings of the present research demonstrate that the metagenome could be used as a biological marker of origin.

## Figures and Tables

**Figure 1 foods-11-02483-f001:**
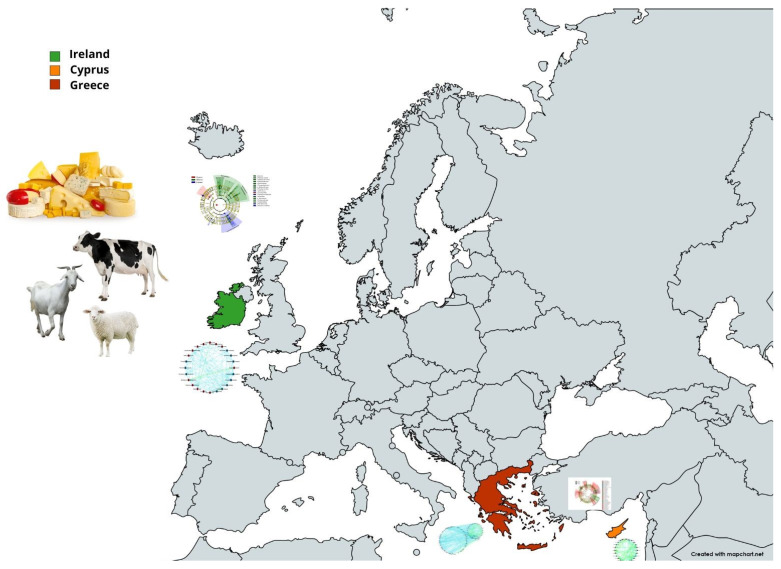
Map of cheese samples collection countries.

**Figure 2 foods-11-02483-f002:**
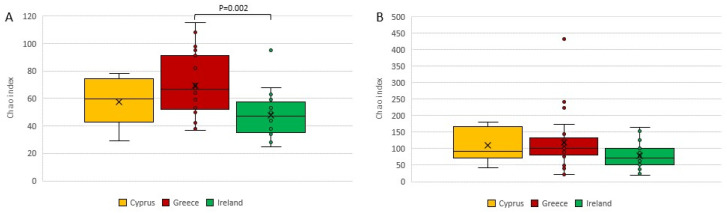
Comparison of (**A**) bacterial and (**B**) fungal alpha diversities of Cyprus, Greek, and Irish cheeses, based on the Chao1 index. Statistical analysis was performed using the Kruskal–Wallis test.

**Figure 3 foods-11-02483-f003:**
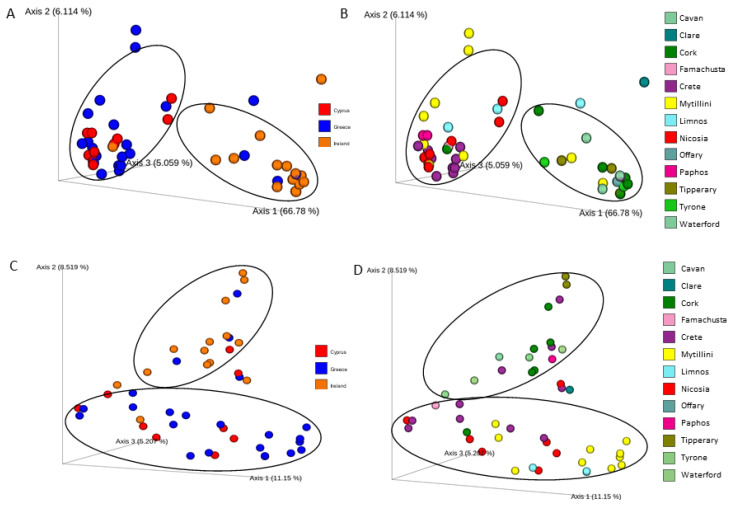
Principal coordinate analysis (PCoA) plot, showing the similarities in microbial beta diversity among cheeses: (**A**) from different countries, regarding bacterial beta diversity; (**B**) from different areas, regarding bacterial beta diversity; (**C**) produced by milk from different animals, regarding bacterial beta diversity; (**D**) from different countries, regarding fungal beta diversity; according to Unweighted Unifrac distance.

**Figure 4 foods-11-02483-f004:**
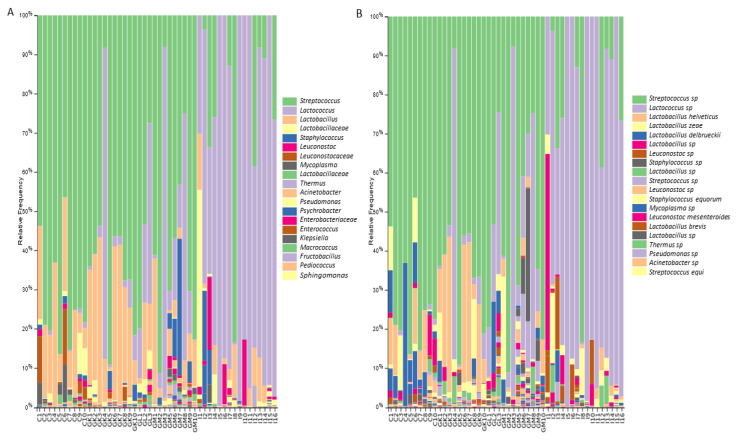
The relative abundance of the twenty most abundant bacteria identified at: (**A**) the genus level; (**B**) the species level, based on 16S rRNA gene sequencing for Cyprus, Greek, and Irish cheeses.

**Figure 5 foods-11-02483-f005:**
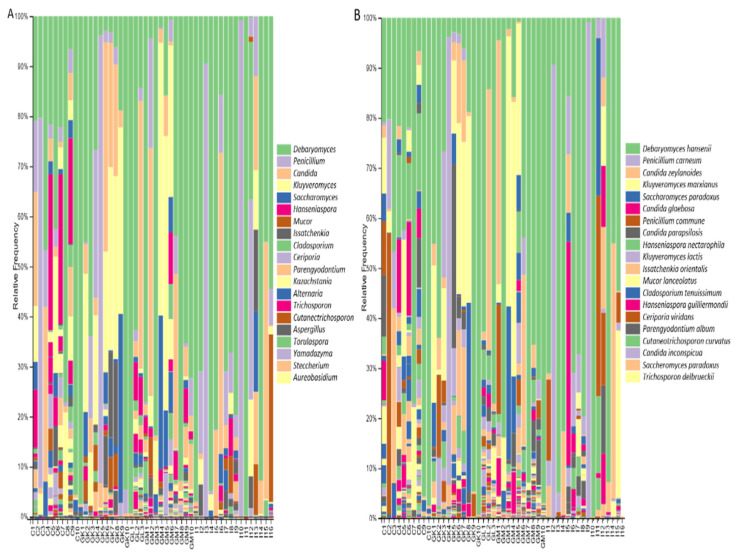
The relative abundance of the twenty most abundant fungi identified at: (**A**) the genus level; (**B**) the species level, based on ITS loci sequencing for Cyprus, Greek, and Irish cheeses.

**Figure 6 foods-11-02483-f006:**
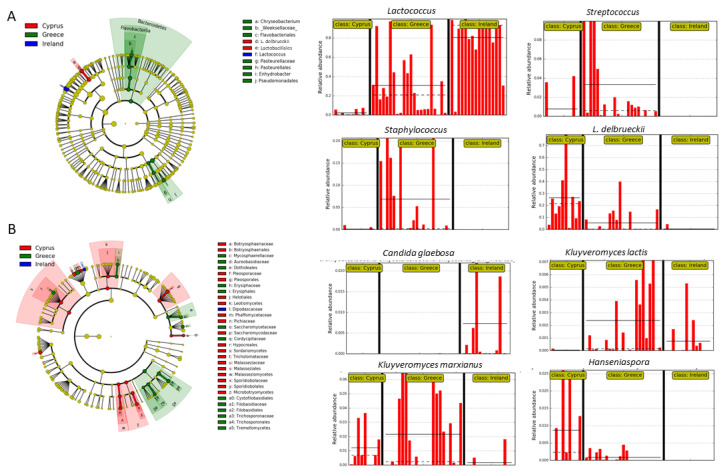
Biomarkers discovery using the LEfSe algorithm showing (**A**) bacterial and (**B**) fungal taxa that indicated statistically significant over-representation among the Irish, Greek, and Cypriot cheeses, based on a non-parametric factorial Kruskal–Wallis (KW) sum–rank test, an (unpaired) Wilcoxon rank–sum test, and LDA. The left part of the picture depicts phylogenetic trees that map the taxonomic differences of the detected biomarkers from class (outside part of the circle) to species level (inside part of the circle), combined with a list of taxa with significantly increased representation in cheeses from Cyprus (red color), Greece (green color), and Ireland (blue color). The right part of the picture shows histograms of the taxa that indicated significantly increased relative representation in cheeses from Ireland, Greece or Cyprus, their detected relative abundances, and in how many samples they were detected.

**Figure 7 foods-11-02483-f007:**
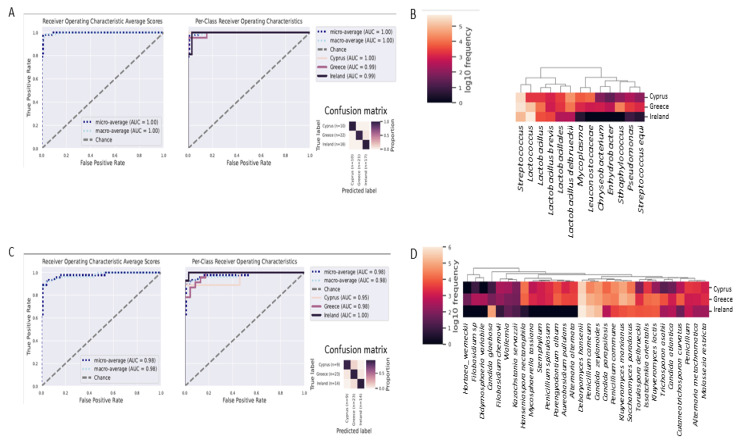
Microbial taxa that can accurately differentiate cheeses from Ireland, Greek and Cyprus. (**A**) ROS and PCROC analyses for bacterial taxa, indicating true and false positive rates for cheeses from each country, shows perfect predictive accuracy (AUC = 1.00). The confusion matrix shows the proportion of how many times every sample gets the correct classification. (**B**) Heatmap showing the 13 top bacterial predictive features that differentiate cheeses from each country. (**C**) ROS and PCROC analyses for fungal taxa, indicating true and false positive rates for cheeses from each country, shows perfect predictive accuracy (AUC = 1.00). The confusion matrix shows the proportion of how many times every sample gets the correct classification. (**D**) Heatmap showing the 30 top fungal predictive features that differentiate cheeses from each country.

**Figure 8 foods-11-02483-f008:**
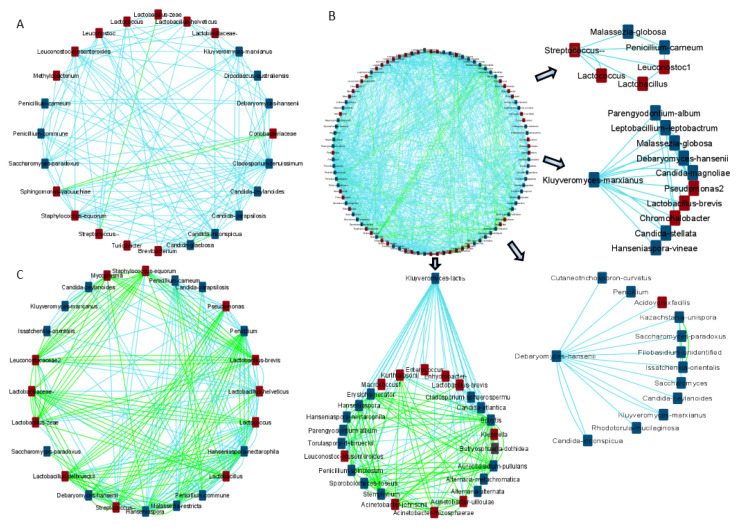
The correlation pattern of microbial communities in: (**A**) Irish cheeses; (**B**) Greek cheeses; (**C**) Cypriot cheeses. Green and light-blue edges represent positive or negative correlations between two nodes, based on Spearman’s rank correlation, the Pearson correlation, and the Bray Curtis and Kullback–Leibler dissimilarity matrices. Red and blue squares represent the bacterial and the fungal taxa, respectively.

**Table 1 foods-11-02483-t001:** Information about the collected samples (cheeses have been anonymized for commercial reasons).

Sample-id	Country of Origin	Area of Production	Type of Cheese	Milk Source	Raw/Pasteurized Milk	Starter Culture	Ripening	Smoking
C1	Cyprus	Nicosia	Kefalotyri	sheep and goat	Pasteurized	lactic acid bacteria	semi	no
C2	Cyprus	Nicosia	Kefalotyri	sheep and goat	Pasteurized	lactic acid bacteria	semi	no
C3	Cyprus	Famachusta	Kefalotyri	sheep and goat	Pasteurized	lactic acid bacteria	hard	no
C4	Cyprus	Nicosia	Kefalotyri	sheep	Pasteurized	lactic acid bacteria	hard	no
C5	Cyprus	Nicosia	Kefalotyri	sheep and goat	Pasteurized	lactic acid bacteria	hard	no
C6	Cyprus	Nicosia	Kefalotyri	sheep and goat	Pasteurized	lactic acid bacteria	semi	no
C7	Cyprus	Nicosia	Kefalotyri	sheep	Pasteurized	lactic acid bacteria	semi	no
C8	Cyprus	Nicosia	Kefalotyri	sheep and goat	Pasteurized	lactic acid bacteria	hard	no
C9	Cyprus	Paphos	Kefalotyri	sheep	Pasteurized	lactic acid bacteria	hard	no
C10	Cyprus	Paphos	Kefalotyri	sheep	Pasteurized	lactic acid bacteria	hard	no
GK1	Greece	Crete	Graviera	sheep and goat	Pasteurized	lactic acid bacteria	hard	no
GK2	Greece	Crete	Graviera	sheep and goat	Pasteurized	lactic acid bacteria	hard	no
GK3	Greece	Crete	Graviera	sheep and goat	Pasteurized	lactic acid bacteria	hard	no
GK4	Greece	Crete	Graviera	sheep and goat	Pasteurized	lactic acid bacteria	hard	no
GK5	Greece	Crete	Graviera	sheep and goat	Pasteurized	lactic acid bacteria	hard	no
GK6	Greece	Crete	Graviera	sheep and goat	Pasteurized	lactic acid bacteria	hard	no
GK7	Greece	Crete	Graviera	sheep	Pasteurized	lactic acid bacteria	hard	no
GK8	Greece	Crete	Graviera	sheep and goat	Pasteurized	lactic acid bacteria	hard	no
GK9	Greece	Crete	Graviera	sheep and goat	Pasteurized	lactic acid bacteria	hard	no
GK10	Greece	Crete	Graviera	sheep	Pasteurized	lactic acid bacteria	hard	no
GL1	Greece	Limnos	Kaskavalli	sheep and goat	Pasteurized	lactic acid bacteria	semi	no
GL2	Greece	Limnos	Kalathaki	sheep and goat	Pasteurized	lactic acid bacteria	semi	no
GL3	Greece	Limnos	Melichloro	sheep and goat	Pasteurized	lactic acid bacteria	hard	no
GM1	Greece	Mytillini	Kefalotiri	sheep and goat	Pasteurized	lactic acid bacteria	hard	no
GM2	Greece	Mytillini	Ladotyri	sheep and goat	Pasteurized	lactic acid bacteria	semi	no
GM3	Greece	Mytillini	Kaseri	sheep and goat	Pasteurized	lactic acid bacteria	semi	no
GM4	Greece	Mytillini	Graviera	sheep	Pasteurized	lactic acid bacteria	hard	no
GM5	Greece	Mytillini	Ladotyri	sheep and goat	Pasteurized	lactic acid bacteria	hard	no
GM6	Greece	Mytillini	Ladotyri	sheep	Pasteurized	lactic acid bacteria	hard	no
GM7	Greece	Mytillini	Ladotyri	sheep and goat	Pasteurized	lactic acid bacteria	hard	no
GM8	Greece	Mytillini	Ladotyri	sheep and goat	Pasteurized	lactic acid bacteria	hard	no
GM9	Greece	Mytillini	Graviera	sheep and goat	Pasteurized	lactic acid bacteria	hard	no
GM10	Greece	Mytillini	Ladotyri	sheep and goat	Pasteurized	lactic acid bacteria	hard	no
I1	Ireland	Clare	Cheddar	goat	Raw	no	semi	no
I2	Ireland	Tipperary	Blue cheese	sheep	Pasteurized	lactic acid bacteria and *Penicillum roquefortii*	semi	no
I3	Ireland	Cork	Artisan	cow	Pasteurized	Starter culture	semi	no
I4	Ireland	Cavan	Artisan	goat, sheep and cow	Raw	Starter culture	hard	no
I5	Ireland	Cork	Artisan	goat	Pasteurized	lactic acid bacteria	hard	yes
I6	Ireland	Cork	Artisan	cow	Pasteurized	Starter culture	semi	yes
I7	Ireland	Cork	Artisan	cow	Pasteurized	Starter culture	hard	no
I8	Ireland	Cork	Fine Swiss	cow	Raw	Starter cultures	hard	no
I9	Ireland	Tipperary	Blue cheese	cow	Pasteurized	starter cultures and *Penicillum roquefortii*	semi	no
I10	Ireland	Cork	Artisan	cow	Pasteurized	Starter cultures	semi	no
I11	Ireland	Waterford	Cheddar	cow	Pasteurized	Starter cultures	hard	no
I12	Ireland	Waterford	Cheddar	cow	Pasteurized	Starter cultures	hard	no
I13	Ireland	Waterford	Cheddar	cow	Raw	Starter cultures	hard	no
I14	Ireland	Waterford	Cheddar	cow	Raw	Starter cultures	hard	no
I15	Ireland	Offaly	Cheddar	cow	Pasteurized	Starter cultures	hard	no
I16	Ireland	Tipperary	Artisan	sheep	Pasteurized	Starter cultures	hard	no

## Data Availability

Data are available at https://dataview.ncbi.nlm.nih.gov/object/PRJNA838921.

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
