# Peer review of "Metataxonomic Mapping of the Microbial Diversity of Irish and Eastern Mediterranean Cheeses"

_foods, 2022, doi:10.3390/foods11162483_

Round 1
Reviewer 1 Report
The article is very interesting and the application of the metataxonomic approach sounds brilliant.
The experimental design is adequate and the conclusions fully supported by the results. All the steps of the research are adequately descripted and supported by graphs and proper statidtical elaboration.
Author Response
We would like to thank the reviewer for reviewing our manuscript. Additionally, we thank the reviewer for acknowledging the significance of our study.

Reviewer 2 Report
The work shows the biological diversity in different types of cheese and the possibility of using biomarkers.
Author Response
We would like to thank the reviewer for reviewing our manuscript. Additionally, we thank the reviewer for acknowledging the importance of our study in cheese microbiota analysis.

Reviewer 3 Report
The paper under consideration intends to evaluate the microbial diversity in cheese from three regions and relate it to the product geographical, a very interesting topic, especially in cases where the authenticity of the origin must be ensured.
It is based on recent methodologies, at the microbial and on the analysis of results level, with very good prospects of interesting results. In my opinion, the sampling offers some fragility on the subject for two main reasons: on the one hand, it ensures cheeses from the three origins but includes a large majority of cheeses made from pasteurized milk, which presupposes dominance of strains from starters used in the manufacture which tend to reproduce the effect of these starters, as the authors refer in the conclusions, and cheeses are more dependent on the process and manufacturing options than on the origin of the cheeses; on the other hand, as can be deduced from the list of cheeses presented and from the results, there are few cheeses (2) whose typology is summarily indicated (the case of Penicillium Roqueforti as a starter) but nothing is known about the other samples except that it is used "starter culture”, which does not define the technological type of cheese which has an influence on the microbial evolution that can be verified during the curing process. Furthermore, I consider that it is a limited sampling when the determining factors are several - type of milk (animal species), milk treatment (raw, pasteurized), type of starter, type of cheese, which can lead to errors of interpretation when it is intended to generalize the results to the comparison between regions. Finally, it would be interesting to give preponderance to traditional raw milk cheeses, which might represent best the influence of origin.
Despite this, the results are well presented and discussed, although the figures show very small characters that make it difficult to read and appreciate. I think this is an important aspect to review.
Finally, I consider that the discussion is extensive but not very objective and concrete regarding the objectives of the work, which justifies some revision of this chapter.
These are the most relevant generic comments about the submitted paper. Detailed suggestions are as follows
Specific suggestions:
Table 1 – In this table we don’t have we did not find elements sufficiently characterizing the technological type of cheese
Figure 3 – I couldn’t find figures 3E and 3F
Figures 4 and following – difficult to read, should be corrected to larger letters
Lines 283 – > or < ?
Lines 524-525 - Shouldn't measures also be considered to prevent contamination during the manufacturing process?
Author Response
We would like to thank the reviewer for reviewing our manuscript. Additionally, we thank the reviewer
for the considerable comments and recommendations, that will assist to the improvement of the
manuscript.
We agree with the reviewer that “the use of pasteurized milk presupposes dominance of strains from
starters used in the manufacture which tend to reproduce the effect of these starters”. However, our
sampling was random and revealed that almost all producers apply pasteurization and the use of starter
cultures as a common practice. Therefore, we analyzed our data based on what is currently applied as
manufacturing practice from the industries, including a) pasteurization and b) the use of starter cultures.
It is important to emphasize that we didn’t have any indication on which starter cultures were applied, or
the manufacturing conditions used as these were purchased commercial cheeses. Also, apart from the
dominant bacteria, we identified fungal contaminants that were distinct signatures for each geographic
location. Furthermore, the study revealed distinct microbial co-abundances that aligned to geographical
location in many instances.
We agree with the reviewer that “factors are several - type of milk (animal species), milk treatment (raw,
pasteurized), type of starter, type of cheese” would have a major influence on the development of the
cheese microbiota. However, instead this analysis was almost agnostic to these factors to determine if
the background metagenome of cheeses could be used as a marker of their origin. Indeed, the application
of machine learning algorithms using random forest revealed that the microbiota can identify the origin
of the cheeses with 100% accuracy regarding bacteria and 98% regarding fungus, rejecting the null
hypothesis.
We agree with the reviewer that “it would be interesting to give preponderance to traditional raw milk
cheeses, which might represent best the influence of origin”. We added it at the last paragraph of the
discussion session, in the future plans.
Regarding the comment that figures show very small characters that make it difficult to read and
appreciate, we had to compress the Figures to fit to the space of the draft manuscript. We also provided
bigger, high-resolution figures, and the size of the current figures in the manuscript was increased.
Please find the corrections in the revised manuscript (with track changes) and our comments highlighted
with red.

Reviewer 4 Report
Dear authors, here are my comments on your work "metataxonomic mapping of the microbial diversity of Irish and eastern Mediterranean cheeses." The authors hypothesized that the background metagenome of cheeses could be used as a marker of their origin. The introduction, materials, methods, results, and discussion are well written and easy to follow.
My main concern is about the conceptualization of this work. First, you describe the microbial biodiversity of many kinds of cheese from diverse country origins. However, 98% of such samples came from pasteurized cheeses (Table 1). Therefore, the microbial biodiversity only shows each country's preference for using certain brands of microbial cultures for cheesemaking. I can support this observation with your figure 4. In this figure, you can observe the high prevalence of commercial bacteria such as Streptococcus, lactococcus, and lactobacillus in most of your samples. The "native" microorganisms represent a tiny proportion in most cheeses, influencing your PCoA.
I would like to hear your comments, mainly because you probably are misinterpreting your data, and your conclusion may be wrong.
Author Response
We would like to thank the reviewer for reviewing our manuscript.
The main aim of the paper was to see if we could differentiate between cheeses of different origins based
on microbial signatures/patterns irrespective of starters or manufacturing characteristics. Our sampling
was random and revealed that almost all producers apply pasteurization and the use of starter cultures
as a common practice. Therefore, we analyzed our data based on what is currently applied as
manufacturing practice from the industries, including a) pasteurization and b) the use of starter cultures.
It is important to emphasize that we didn’t have any indication on which starter cultures were applied, or
the manufacturing conditions used as these were purchased commercial cheeses.
We agree with the reviewer in one point, that the dominant bacteria (not fungi) might indicate each
country's preference for using certain brands of microbial cultures for cheesemaking. The microbial
readout would reflect starters used and we discuss this, but would also come from dead bacteria, nonstarter lactic acid bacteria, post pasteurization/plant contaminants and fungi/yeasts. The paper
demonstrates through co-abundance analysis that it is possible to differentiate the cheeses based on
fungi/yeast profiles and non-starter bacteria. Indeed, the application of machine learning algorithms using
random forest revealed that the fungal diversity could identify the origin of the cheeses with 98%
accuracy, rejecting the null hypothesis. Additionally, the study revealed distinct microbial co-abundances
that aligned to geographical location in many instances.
